# Oral Health Service Access in Racial/Ethnic Minority Neighborhoods: A Geospatial Analysis in Washington, DC, USA

**DOI:** 10.3390/ijerph19094988

**Published:** 2022-04-20

**Authors:** Meirong Liu, Dennis Kao, Xinbin Gu, Whittni Holland, Gail Cherry-Peppers

**Affiliations:** 1School of Social Work, Howard University, Washington, DC 20059, USA; whittni.holland@bison.howard.edu; 2School of Social Work, Carleton University, Ottawa, ON K1S 5B6, Canada; denniskao@cunet.carleton.ca; 3College of Dentistry, Howard University, Washington, DC 20059, USA; xgu@howard.edu (X.G.); gail.cherrypeppers@Howard.edu (G.C.-P.)

**Keywords:** spatial accessibility, oral health services, racial and ethnic neighborhoods, African Americans

## Abstract

Previous studies on individual-level variables have improved our knowledge base of oral health service use. However, environmental or contextual variables are also important in understanding oral health disparities in racial and ethnic neighborhoods. Based on Bronfenbrenner’s ecological framework, this study examines the geographic availability of oral health providers in Washing-ton DC, U.S.A. Census tract-level data were drawn from the American Community Survey, joined with tract-level shapefiles, and overlaid with the geographic location of dental services throughout the city. Visual maps, descriptive statistics, and spatial lag regression models showed that census tracts with higher concentrations of African Americans were significantly farther from their nearest oral health providers (r = 0.19, *p* < 0.001), after controlling for neighborhood poverty rate, median age, and gender. Such findings confirm that in urban areas with highly di-verse populations such as Washington DC, racial disparities in oral health care access are signifi-cant. The study highlights that identifying neighborhoods with limited oral health care providers should be a priority in diminishing racial disparities in oral health service access. Improving access to racial/ethnic minority communities, especially African American neighborhoods, will require changes in health policies and programs, workforce development, resource allocation, community outreach, and educational programs.

## 1. Introduction

The importance of oral health as a component of comprehensive health has been well documented. Regular preventive oral health care can detect problems early when they are usually easier to treat. However, many people do not get the care they need, due to inaccessibility of quality services. Untreated oral health problems can lead to difficulties in chewing food, pain, systemic infections, hospitalization, and in some cases, death [1,2]. Oral health is directly associated with different disease conditions and is closely linked to general health [3,4]. Dental diseases, such as tooth decay and gum disease, are linked to chronic health conditions, including poor nutrition, stroke, coronary heart diseases, diabetes, obesity, and kidney disease [5,6]. Locoregional infections of dental origin have the potential to cause sepsis [7]. Poor oral health can also result in lost work hours, functional limitation, depression, and poor quality of life [8].

The Healthy People 2030 initiative highlights the need to reduce tooth decay and other oral health conditions and support access to oral health care services [9]. Progress has been made over the past years in identifying specific vulnerable groups as it relates to their access to oral health. However, despite significant advances in oral health sciences, including the development of preventive, diagnostic, and therapeutic agents and methods, oral health disparities persist [10]. Research has documented the existence of disparities in oral health across a broad range of sociodemographic factors (e.g., race or ethnicity, gender, education, income, disability, geographic location, and sexual orientation) [11], as well as social, environmental, behavioral, cultural, economic, and political factors [12]. Socioeconomically disadvantaged populations have the highest levels of untreated dental disease because of a lack of quality care [13]. People with oral health deterioration are more likely to be low-income, have no insurance, and members of racial/ethnic minority populations [14].

Not surprisingly, major racial disparities exist in oral health in the United States [15]. Lower rates of dental visits and preventative care contribute to the high rates of untreated tooth decay among African Americans [16,17]. African American adults are nearly twice as likely to have untreated decay compared to their white counterparts [18]. With higher rates of untreated decay, racial minority adults are more likely to lose their teeth due to dental disease. African American adults are at greater risk of having a tooth extracted due to tooth decay or gum disease, and seniors are much more likely to have lost all of their teeth [18]. They also fare worse overall and with regards to disease-specific survival rates for oral and oropharyngeal cancers [19].

The lack of accessible and/or quality services is a contributing factor to the existing racial/ethnic disparities. African Americans reported higher unmet dental care needs than their white peers (66.3% vs. 40.2%) [20]. Racism is prevalent throughout the health system. Implicit bias arising from subconscious thoughts or beliefs may also influence dental care providers’ approach, decision making, or patient management [21]. For instance, Patel et al. [22] found that a dentist’s decision making was affected by the race of the patient, resulting a greater likelihood of extractions for African American patients with a broken-down tooth and symptoms of irreversible pulpitis.

While individual-level characteristics are important, environmental or contextual variables can also influence oral health disparities. At the macro level, oral health can be affected by various neighborhood factors, including physical safety, culture, social networking, oral health information, and the supply of dental care providers [23]. Research suggests that geography is an important consideration. For example, distinct oral health disparities and inequities exist among those residing in medically and dentally underserved rural and urban areas. Saman et al. [24] applied geospatial illustration methods to show the effects of one’s place of residence in understanding oral health disparities in Kentucky. Kurcz et al. [25] used the Geographic Information Systems (GIS) tools to map private dental practices in the State of Indiana and found mean personal income by place of residence did not significantly influence practice location. Brown et al. [26] found that adults living near a neighborhood with resources were less likely to report fair or poor oral health. Finally, Yoon et al. [27] found that Asian Americans in Texas living in areas with a higher level of available dentists were more likely to use preventive dental care services, emphasizing the importance of the location (proximity or accessibility) to dental clinics.

## 2. The Setting: Washington, DC

This study focused on the city of Washington DC, which is the national capital of the United States, with a population of approximately 690 thousand residents. The city is subdivided into eight smaller governmental units called wards. The Anacostia River divides the city not just naturally but economically and racially, clustering wealth and resources disproportionately west of the river, and poverty east of the river, with sharply different consequences for the lives of the residents [28]. Wards 1 through 6 are located west of the Anacostia River, and Wards 7 and 8 are located east of the river.

As in many U.S. urban cities, the city is racially segregated. Wards 2 and 3 in the northwest quadrant are largely made up of white people, whereas Wards 7 and 8 are largely African Americans. Wards 7 and 8 have long been underserved, with resources disproportionately invested to the west of the river [29]. Many environmental factors that are associated with poor health outcomes, including experiences with discrimination, violence, reduced access to health services, and poverty, disproportionately affect individuals in these Wards [30,31]. Unfortunately, factors related to oral health treatment utilization specific to racial and ethnic minority neighborhoods, e.g., Wards 7 and 8, have previously gone unexamined [32].

Bronfenbrenner’s Ecological Systems Framework situates individuals in the context of their environment, in which the interactions between multiple systems can influence individual health, well-being, and behaviors [33,34]. Under this framework, access to care is not only influenced by personal characteristics but broader contextual factors, including one’s environment. Specifically, this study relies on the concept of spatial accessibility, which involves the geographic location of services as it relates to where people are situated. Spatial accessibility is based on the geographic notion of “distance decay” and assumes that the closer the services are geographically located, the more likely an individual is to utilize those services. Based on this concept, we aim to examine residents’ access to oral health providers in the city, asking the research question: Is the spatial accessibility of oral health services less for racial/ethnic minority communities? We first examine the spatial distribution of oral health services in the city. Then, we investigate how neighborhood factors, including neighborhood racial composition and poverty rates, might be related to the residents’ access to oral health services.

## 3. Methods

### 3.1. Data and Measures

Geographic and demographic data were drawn from various sources. We utilized a shapefile of tracts based on 2018 designated boundaries from the US Census Bureau [35]. Tract-level neighborhood demographic characteristics including race/ethnicity (African American and Hispanic), median age, poverty rates, and gender were drawn from the 2018 five-year American Community Survey [36]. Data on oral health services were purchased from dentistlistpro.com, accessed on 8 February 2021, a dentist database that contains locations of private oral health care providers. We also included the locations of the Federally Qualified Health Centers (FQHCs), i.e., community-based health care providers that provide oral health services to meet the needs of underserved areas [37]. The DC Ward boundaries were obtained from Open Data DC, a DC government database site [38].

### 3.2. Methods of Analysis

The analysis consisted of three primary stages. First, maps and descriptive statistics were conducted using QGIS (https://qgis.org, accessed on 28 April 2021) [39]. QGIS is an open-source professional GIS application that allows one to map together and analyze different map layers and data. The resulting maps have several layers which represent different features, including the location of oral health providers, census tracts, ward boundaries, and other physical features included for geographic reference, e.g., major streets and rivers. The tract-level neighborhood demographic data were joined to the tract shapefile, allowing us to create thematic or choropleth maps and use continuous color palettes (e.g., lighter blue to darker green) to highlight varying demographic characteristics (e.g., percent African American).

Second, the Pearson’s correlations between the neighborhood tract-level demographic characteristics and the distance to the nearest oral health provider were computed using Tableau (Seattle, WA, USA), an interactive data visualization software. To measure the spatial accessibility of oral health provides, we utilized GIS technology (ArcGIS Online, Esri, Redlands, CA, USA) to calculate the distance in miles from the geographic centroid of each tract to the nearest oral health provider. Specifically, we employed the “Find Nearest” tool, which measures the distance from any point of one feature to the nearest point of another feature, utilizing the shortest route along any combination of roads. Because the census tracts are represented as polygons, we calculated the geographic center (centroids) to approximate the location of each tract. The routes were then calculated from each census tract centroid to the nearest oral health provider.

In the final stage, spatial lag regression modeling was estimated using GeoDa https://geodacenter.github.io, accessed on 17 April 2022). Unlike ordinary least squares (OLS) regression models, spatial lag regression models account for violations of the uncorrelated error terms and independent observations assumptions [40], which is common in working with spatial data. Spatial lag models assume that the dependent variable for any given place (census tract) is affected by any neighboring units, or in other words, is more similar to geographies which are closer to each other. The likelihood of such spatial autocorrelations was thus controlled for in the models.

## 4. Results

Figure 1 presents the map of 755 oral health providers and 26 FQHCs in Washington DC, overlaid with the ward boundaries. Visually, it appears that several wards (i.e., Wards 5, 7, and 8), have fewer overall dental services. Most of the FQHCs also appear to be in these wards.

These observations are confirmed in Table 1, which breaks down the number of oral health providers by Ward. More than half of the oral health providers (54.1%) were located in Wards 2 and 3 (highlighted in dark blue), with almost a third of all services located in Ward 2. In terms of neighborhood characteristics, Ward 2 had only 13.2% African American residents and a poverty rate of 5.9%; Ward 3 had 5.3% African American residents and only 2.3% of the residents were living in poverty. In contrast, only 12.5% of all oral health providers were in Wards 5, 7, and 8 in total, with Ward 8 having the fewest providers. These three wards—but particularly Wards 7 and 8 (highlighted in light blue) —were also home to significant populations of African American and poor families. The majority of FQHCs (73.1%) were also located in these three communities, with 30% of FQHCs located in Ward 8, 26.9% located in Ward 7, and 15.4% in Ward 5, respectively.

The maps in Figure 2 further confirm these patterns at the census tract level. The tracts with the highest concentration of African Americans tend to be in Wards 5, 7, and 8. Poverty is a bit more dispersed across the city but tends to also be concentrated in these three southern wards. The Hispanic/Latino communities seem to be most concentrated in Wards 1 and 4, as well as Ward 3.

Table 2 summarizes the correlations between the census tract-level demographic characteristics (i.e., percentage of African Americans, Hispanics, male, poverty, and median age) and the spatial accessibility of oral health (i.e., the distance to the nearest oral health providers).

We first computed the correlations between the neighborhood characteristics with only private oral health providers. We then added the FQHCs to examine if their presence helped to buffer the associations between neighborhood characteristics and residents’ oral health care access. The results indicated that residents in tracts with higher concentrations of African Americans had significantly farther distances to the nearest oral health providers (i.e., less access, r = 0.26, *p* < 0.0001). The addition of FQHCs seemed to diminish this association a bit (r = 0.19, *p* < 0.001), but there was still significantly less oral care access in tracts with higher percentages of African Americans. Residents in tracts with higher poverty rates also had significantly less access (r = 0.17, *p* < 0.05), but with the addition of FQHCs, the correlation was no longer significant. Tracts with a higher percentage of Hispanic residents and higher median ages were associated with greater spatial accessibility. Residents in neighborhoods with more males did not have a farther distance to nearest oral health providers, but this correlation became significant with the addition of FQHCs (r = 0.11, *p* < 0.05). Figure 3 provides a visual of the relationship between poverty rates and access to oral health providers. Tracts with a higher percentage of African American population were associated with less oral health care access, but this association decreased with the addition of FQHCs.

Finally, Table 3 presents the regression results of the OLS and spatial lag models on distance to the nearest oral health providers, controlling for the tract-level neighborhood characteristics. The OLS regression results are shown for comparison purposes. The spatial lag effects were significant, showing strong similarities among neighboring census tracts, and thus, highlighting the importance of accounting for the spatial autocorrelation. Overall, the models confirmed that after controlling age, gender, and poverty, tracts with higher concentrations of African Americans were still associated with farther distances to oral health providers or less spatial access (β = 0.28, *p* < 0.01). This was also the case with the addition of FQHCs, but the effect seemed to be smaller (β = 0.23, *p* < 0.05). In contrast, tracts with more Hispanic persons were associated with decreased distances to oral health providers (β = −0.61, *p* < 0.05). Further, the results suggest that census tracts with higher male populations were also associated with decreased oral health care access (β = 1.46, *p* < 0.001). In addition, the tract-level median age was negatively related to the distance to the nearest dental offices (i.e., residents in a census tract with higher median age had a farther distance to the nearest dental offices) (β = 0.01, *p* < 0.01)), and this association persisted even with the additions of FQHCs (β = 0.01, *p* < 0.05). The regression results showed that after controlling for race, age, and gender, tract-level poverty rate was not associated with the distance to the nearest oral health care providers.

## 5. Discussion

Our findings confirmed that in Washington DC, neighborhood-level demographic characteristics—particularly the presence of racial/ethnic minorities—have important implications for the oral health access of community residents. Compared to the other wards, Wards 7 and 8 have the highest percentages of African American population and highest poverty rates, as well as a shortage of oral health care providers [32]. Our models showed that neighborhoods with high percentages of African Americans had less spatial accessibility, even after controlling for neighborhood poverty rate, median age, gender, and accounting for the presence of FQHCs.

Such findings are consistent with previous findings. For instance, in 2014, DC Health [41] found that while 72.5% of the DC residents had reported having a dental visit, there were significant geographic discrepancies, with rates as low as 59.7% in Ward 7 and 64.4% in Ward 8. Moreover, 16.1% of African Americans versus 2.2% of whites had six or more permanent teeth removed in DC [41]. Residents living in Wards 7 and 8 also reported higher rates of health problems compared with the rest of the city [42]. The lack of spatially accessible oral health care provides one potential explanation.

Our findings are also consistent with previous research showing that a spatially unequal distribution of dentists or dental care professionals could reduce the quality of health services and increase health inequities (e.g., [43]). Our study also showed that neighborhoods with higher concentrations of males have decreased access to oral health care. African American men are especially significantly less likely to use dental services [16,44] and are among the most disadvantaged racial/gender cohorts with respect to oral health access [16,45].

## 6. Implications

Racial health inequity is a multifaceted concern, and the accessibility of services is one important dimension [46]. Examination of the distribution of oral health care providers practicing across DC reveals inequity in oral health care access [32]. The findings highlight the need for policymakers, researchers, and providers to expand the focus from individual-level factors to broader contextual processes. Systemic racism is at the root of racial inequalities in oral health, and related disparities cannot be effectively reduced by only addressing individual-level factors [15]. Thus, multilevel interventions should be utilized to address the individual, contextual, and social determinants of oral health access and meet the needs of underserved minority communities [16].

It is also critical to continue developing integrative technologies and health care models that account for the needs of individuals and families in racial and ethnic communities with limited resources. Federal agencies (such as the Health Resources and Service Administration, the National Institute of Dental and Craniofacial Research, and the Agency for Healthcare Research and Quality)—in partnership with community health organizations—should prioritize programs, research, and health policies that seek to address oral health care equity [47]. For example, the Institute of Medicine and the National Research Council have recommended that the HRSA further expand the capacity of FQHCs to deliver essential oral health services [48]. The current study provides support for the important role of FQHCs in addressing access issues in disadvantaged racial and ethnic neighborhoods. FQHCs offer a wide range of services in low-income or underserved neighborhoods and form long-lasting, trusting relationships with the underserved communities. They are deeply embedded in the community and uniquely positioned to provide integrated and patient-centered care for low-income populations. FQHCs can employ a variety of strategies to integrate oral health and primary care service delivery, such as engaging with primary care clinicians to provide oral health screening and referral services [48].

The recruitment and retention of diverse oral health care providers needs to be a national priority. First, educational programs are needed to attract students with affinity to underserved areas and provide curricula and opportunities for more emerging professionals to serve in underserved areas (e.g., via clinical rotations, compulsory community services, etc.). Innovative programs that are linked to serving in underserved areas, such as student loan repayment, reduced tuitions or scholarships, and other financial incentives, are strongly encouraged [32]. Second, supporting oral health care providers in underserved areas can be an important approach, especially regarding the retention of personnel. Health policies may offer incentives for providers to treat patients who are from underserved communities through the provision of equipment and staff, tax deduction, higher reimbursement rates, or other economic incentives. Other strategies can include providing staffing, the provision of adequate infrastructure for effective workplace organization, housing programs, and the establishment of career pathways. At present, there is no documented evidence of the effectiveness of such measures in oral health; however, valuable insights can be gained when evaluating studies from other medical disciplines [43]. Finally, other innovative programs, e.g., utilizing dental hygienists to perform and promote basic oral hygiene in community settings, may help to respond to the need for oral health services.

There is also a critical need for culturally competent educational and community outreach programs on oral health literacy, attitudes, oral health information, and self-care interventions. Previous studies have shown that lower educational attainment is associated with decreased oral health literacy levels and heightened risks for dental diseases [49,50]. In addition, a lack of awareness around oral health is also a significant barrier [51]. For example, oral health intervention strategies on improving oral health literacy and self-efficacy can be targeted for African American men in underserved neighborhoods. Dragojevic et al. [52] found the success of persuasive health messages depends not only on message content (i.e., what is said), but also on how that content is linguistically framed and delivered (i.e., how it is said). A community-based participatory research approach, which involves community leaders and members, can be adopted when developing effective oral health education programs in racial and socially disadvantaged neighborhoods [46,53]. Dental health educators can play a unique and valuable role. For example, they can collaborate with dental health providers in promoting oral health programs that better support racial and ethnic communities through effective interprofessional, skill-mix efforts [54].

These efforts can be linked with broader public health initiatives. For example, 19.4% of African American adults currently smoke [55]. At the same time, African American men are also at the highest risk of developing oral cancer [16]. Routine dental visits can examine the oral cavity for signs of mucosal and tooth changes, recognize variations from normal, and establish management and referral pathways [56]. Dental providers are also ideally positioned to provide brief interventions to assist their patients to quit smoking [56].

Finally, social support, such as peer support, may be a promising factor that should be explored in further research and practices in addressing oral health utilization for underserved communities. Supports, such as transportation, childcare, and oral health care information and knowledge, can be helpful for racial/ethnic minorities facing barriers to seeking oral health care [45].

## 7. Limitations

The current study has several limitations that can be usefully addressed through future research. First, we used data from one urban area to illustrate the racial disparities in oral health access in the U.S. Although our findings were consistent with other studies’ results on the spatial accessibility of oral health, more research is needed to examine how such disparities are manifested in other cities and regions. Second, it is likely that residents in rural areas face different access issues than those in urban areas, and considerations of access related to travel to oral health providers may be more prominent. Research focusing on residents’ access to preventive checkups and dental treatment in rural areas is also much needed. Third, because of the limitation of the data, this study was not able to reveal other important factors, such as physical safety, social networking, and information regarding oral health care and dental services, that may also impact access to oral health care. Future research should be conducted to further examine how such factors may influence oral health care utilization.

## 8. Conclusions

Although access to oral health care has increased as a result of improvements in insurance coverage and through community-based programs, pronounced oral health disparities still exist. This study confirmed that these challenges are present in Washington DC, but these issues may also be found in other similar urban areas with diverse populations. The situation in Wards 7 and 8 are emblematic of distressed, isolated, and historically disadvantaged communities in cities across the United States. Understanding these oral health disparities is key for developing efforts to facilitate oral health treatment in urban neighborhoods. The findings may help to inform policy makers, community stakeholders, and health care practitioners in the development of multilevel interventions in addressing oral health service accessibility and utilization in racial and ethnic neighborhoods.

## Figures and Tables

**Figure 1 ijerph-19-04988-f001:**
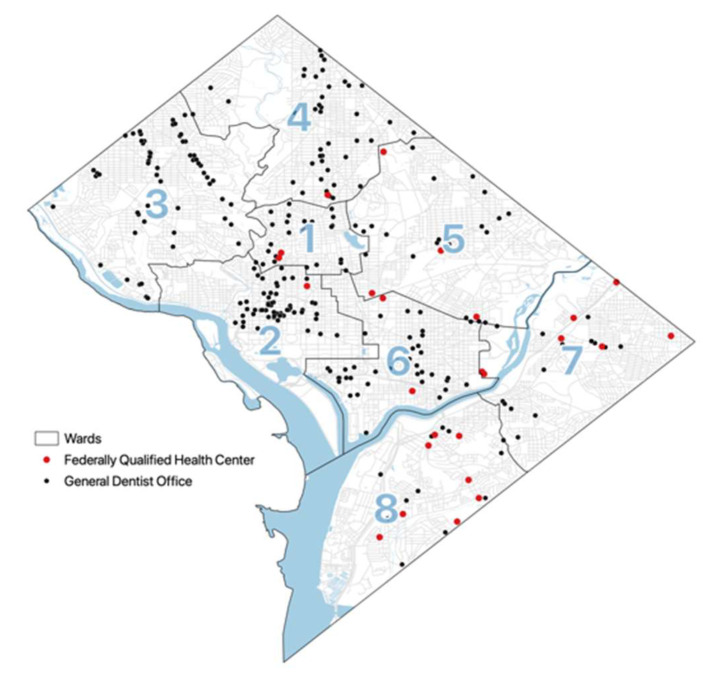
Map of Dental Services in Washington DC (adapted from Davis et al., 2022).

**Figure 2 ijerph-19-04988-f002:**
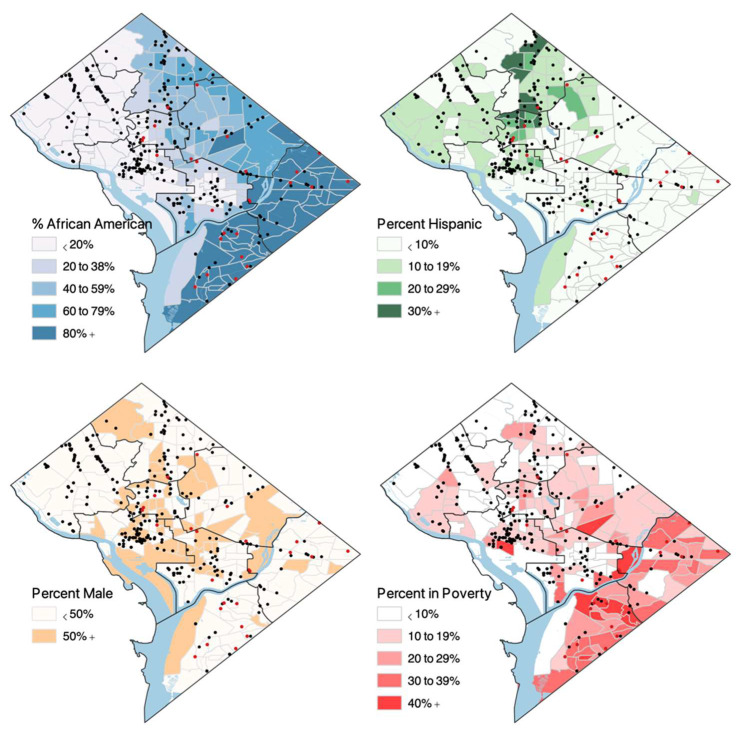
Map of Dental Services & Selected Tract-Level Characteristics.

**Figure 3 ijerph-19-04988-f003:**
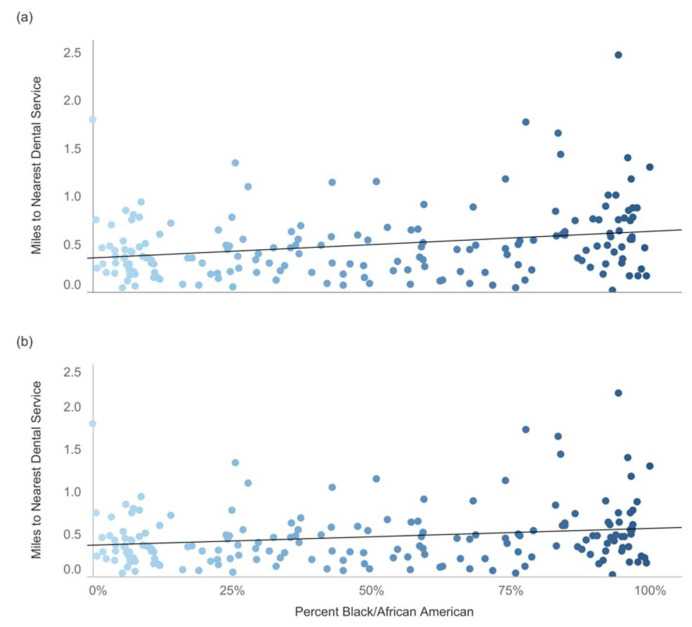
Correlation Between African American Population & Distance to Nearest Dental Service (by Census Tract). (**a**) Dental Services (R-squared = 0.07, *p* < 0.001); (**b**) Dental Services, including Federally Qualified Health Centers (R-squared = 0.04, *p* < 0.01).

**Table 1 ijerph-19-04988-t001:** Dental Services by Ward, Washington DC.

Wards	Dentists	%	FQHCs	%	% African American	% Families in Poverty
All	755	100.0	26	100.0	43.9	12.1
1	74	9.8	3	11.5	21.2	11.9
2	261	34.6	1	3.8	13.2	5.9
3	147	19.5	0	0.0	5.3	2.3
4	97	12.8	1	3.8	45.9	6.8
5	36	4.8	4	15.4	55.4	7.7
6	82	10.9	2	7.7	38.8	7.9
7	27	3.6	7	26.9	91.7	23.3
8	31	4.1	8	30.8	91.8	26.6

**Table 2 ijerph-19-04988-t002:** Correlation coefficients between Tract-Level Demographic Characteristics and Distance to Nearest Dental Office (*n* = 179 census tracts).

Characteristics	Dental Offices	Dental Offices + FQHCs
Percent African American	0.26 ***	0.19 **
Percent Hispanic	−0.30 ***	−0.30 ***
Percent Male	0.11	0.11 *
Percent Poverty	0.17 *	0.17
Median Age	−0.18 *	−0.18

* *p* < 0.05, ** *p* < 0.01, *** *p* < 0.001.

**Table 3 ijerph-19-04988-t003:** Regression Results (*n* = 179 census tracts).

	Dental Offices		Dental Offices + FQHCs	
Characteristics	OLS	Spatial Lag	OLS	Spatial Lag
Percent African American	0.32 **	0.28 **	0.25 *	0.23 *
Percent Hispanic	−1.08 **	−0.61 *	−0.98 **	−0.53
Percent Male	1.36 **	1.46 ***	1.50 **	1.54 ***
Percent Poverty	−0.37	−0.43	−0.33	−0.36
Median Age	−0.01 *	−0.01 **	−0.01 *	−0.01 *
Spatial Lag	-	0.46 ***		0.47 ***
Constant	0.24	−0.04	0.11	−0.14
R-squared	0.19	0.30	0.16	0.29
AIC	123.8	105.7	111.7	91.6

* *p* < 0.05, ** *p* < 0.01, *** *p* < 0.001.

## Data Availability

The data, materials, and code used in this study are available upon request.

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
