# Peer review of "Oral Health Service Access in Racial/Ethnic Minority Neighborhoods: A Geospatial Analysis in Washington, DC, USA"

_ijerph, 2022, doi:10.3390/ijerph19094988_

Round 1

Reviewer 1 Report

The study analyzes the density and availability of dental services in Washington DC and comes to the unsurprising conclusion that there are fewer dentists in low-income neighborhoods inhabited by predominantly African American communities. The paper is clear and raises a number of relevant issues, especially in the discussion. At the same time, there is no doubt that in the U.S. private health insurance system, despite Obama’s reforms, the financial condition of citizens still determines the availability of services. Obviously, emergency dental interventions are provided among those eligible for Medicaid, but quality dentures require serious financial effort. Therefore, I would place little emphasis on racial disparities because the problem is related to the income-social situation.

Author Response

The study analyzes the density and availability of dental services in Washington DC and comes to the unsurprising conclusion that there are fewer dentists in low-income neighborhoods inhabited by predominantly African American communities. The paper is clear and raises a number of relevant issues, especially in the discussion. At the same time, there is no doubt that in the U.S. private health insurance system, despite Obama’s reforms, the financial condition of citizens still determines the availability of services. Obviously, emergency dental interventions are provided among those eligible for Medicaid, but quality dentures require serious financial effort. Therefore, I would place little emphasis on racial disparities because the problem is related to the income-social situation.

Response: Thank you for your positive feedbacks. We really appreciate your valuable suggestions.

In the regression results of the OLS and spatial lag models on neighborhood characteristics and distance to the nearest oral health providers (Table 3), after controlling for poverty, census tract with higher concentration of African Americans are associated with decreased oral health care access (β=0.28, p<0.01), that is farther distances to oral health providers, even with the addition of FQHCs into these neighborhoods (β=0.23, p<0.05). The regression results also showed that after controlling for race, age, and gender, census tract level poverty rate was not associated with distance to the nearest oral health care providers.

Further, in Figure 2 Map of Dental Services & Selected Tract-Level Characteristics, there are neighborhoods that have high percent of poverty but not high percent of African Americans. However, the residents in these neighborhoods have better access to oral health services (Figure 2: Percent in Poverty). These suggest that in our analysis, race/ethnicity is a significant factor for oral health services.

Further, the correlation in Table 2 Correlation coefficients between Tract-Level Demographic Characteristics and Distance to Nearest Dental Office, the correlation between census tract percent poverty and distance to dental services diminished after adding FQHC. But the correlation between percent of African American and distance to dental services is still statistically significant after adding FQHC. Such findings suggest that FQHC addresses the lack of dental service in the neighborhoods with higher percentages of poverty, but not addressing the lack of dental services in the neighborhoods with high percent of African Americans.

There also have been previous literature that racial disparities in oral health access. For example,

The causes of racial disparities in oral health inadequate access to oral health care (Jones et al., 2013); Racial disparities in oral health care access (Davis et al., 2022). Previous literature suggest that systemic racism is the root cause for racial inequalities in oral health, and related disparities cannot be effectively reduced by only addressing micro-level factors (Bastos et al., 2018). Multilevel interventions should be utilized to address the individual, contextual, and social determinants of oral health access that meet the needs of underserved minority communities (Akintobi et al., 2018).

In conclusion, our findings confirmed that in urban areas with highly diverse populations like Washington DC, racial disparities in oral health care access are significant and identifying neighborhoods with limited oral health care providers should be a priority in diminishing racial disparities of oral health service access. Such findings are also consistent with previous literature.

Reviewer 2 Report

Review

The current study - Oral Health Service Access in Racial / Ethnic Minority Neigh- 2 borhoods: A Geospatial Analysis in the District of Columbia - provides support for policy and program efforts towards this direction, especially on expanding the capacity of FQHCs in disadvantaged racial and ethnic neighborhoods.

I think this study is of interest and may sound alarming about the inequalities reflected in the oral health of the African American population.

The article is original and may guide future decision makers of dental organizations to redistribute health services.

I have a few remarks to complete this article, namely:

Abstract

It contains information but should be better structured and supplemented with some data from concrete results.

Introduction

I consider that the term disparities should be replaced by inequalities for example or another term:

For example, line 47 and line 48 - reformulate instead of health disparities in inequalities in access to health services.

Line 45 - reformulated disadvantages with unfavorable socio-economic status.

Line 61 - reformulate communities of color - I suggest a single term of reference for the target population, for example African American people.

Line 66 - I suggest changing the term tooth pulled with tooth extracted.

The line containing lines 69-71 should be reformulated, it is not very well understood.

Material and method

I think that this section could have some improvements, to be clearer.

Result

Table 1 - Rename Black with African American people or another term that you use everywhere

Figure 2 - Rename Percent Black to African American people or another term you use everywhere

Table 2 - Renamed Percent Black

I suggest moving table 2 to Line 189

I suggest that after Figure 3 appears in the text, Figure 3 should also appear.

Figure 3 is not clear why two graphs are the same but with different colors, please explain that it is not understood, possibly explanations on each graph called A and B or otherwise.

Table 3 - line 234 change percent Black with the chosen term.

Discussions - the discussions are comprehensive and to the point

Conclusions - I suggest removing the sentence with the bibliography -line 335 and reformulate.

Bibliography - Bibliographic sources must be entered in numbers with numbers in square brackets according to the MDPI bibliography guide.

Bibliography must be written in MDPI format, see format.

Author Response

Reviewer 2.

The current study - Oral Health Service Access in Racial / Ethnic Minority Neighborhoods: A Geospatial Analysis in the District of Columbia - provides support for policy and program efforts towards this direction, especially on expanding the capacity of FQHCs in disadvantaged racial and ethnic neighborhoods.

I think this study is of interest and may sound alarming about the inequalities reflected in the oral health of the African American population.

The article is original and may guide future decision makers of dental organizations to redistribute health services.

I have a few remarks to complete this article, namely:

Abstract

It contains information but should be better structured and supplemented with some data from concrete results. 

Response: We thank you very much for your time reviewing our study. Your valuable suggestions are greatly appreciated.

We have rewritten the abstract and supplemented with data from concrete results. The abstract has 200 words limit.

Abstract: Previous studies on individual-level variables have improved our knowledge base of oral health service use. However, environmental or contextual variables are also important considerations to understand oral health disparities in racial and ethnic neighborhoods.  Based on Bronfenbrenner’s ecological framework, this study examines the geographic availability of oral health providers in Washington DC, U.S.A. Census tract-level data were drawn from the American Community Survey, joined with tract-level shapefiles, and overlaid with the geographic location of dental services throughout the city. Visualization maps, descriptive statistics, and spatial lag regression models were utilized. The results show that census tracts with higher concentrations of African Americans were significantly farther to their nearest oral health providers (r=0.19, p<0.001), after controlling for neighborhood poverty rate, median age, gender, and even accounting for the addition of Federal Quality Health Centers. Such findings confirm that in urban areas with highly diverse populations such as Washington DC, racial disparities in oral health care access are significant. The study highlights that identifying neighborhoods with limited oral health care providers should be a priority in diminishing racial disparities of oral health service access. Improving access to racial/ethnic minority communities, especially African American neighborhoods, will required changes in health policies and programs, workforce development, resource allocation, community outreach, educational programs.

Introduction

I consider that the term disparities should be replaced by inequalities for example or another term:

For example, line 47 and line 48 - reformulate instead of health disparities in inequalities in access to health services.

Response: Thank you for the comments. Most of the literature we cited use the word “disparities” and therefore, we try to be consistent with existing literature. For examples:

Northridge, M. E., Kumar, A., & Kaur, R. (2020). Disparities in access to oral health care. Annual Review of Public Health, 41, 513–535. https://doi.org/10.1146/annurev-publhealth-040119-094318

Saman, D. M., Johnson, A. O., Arevalo, O., & Odoi, A. (2011). Student column: Geospatially illustrating regional-based oral health disparities in Kentucky. Public Health Reports, 126(4), 612-618.

Lee, J.Y., & Divaris, K. (2014). The ethical imperative of addressing oral health disparities: A unifying framework. Journal of Dental Research, 93(3), 224–230. https://doi.org/10.1177/0022034513511821

Lee, W.C., Li, C.Y., Serag, H., Tabrizi, M., & Kuo, Y.‐F. (2020). Exploring the impact of ACA on rural‐urban disparity in oral health services among US noninstitutionalized adults. Journal of Rural Health, 37(1), 103-113. https://doi.org/10.1111/jrh.12418

Osazuwa-Peters, N., Massa, S. T., Christopher, K. M., Walker, R. J., & Varvares, M. A. (2016). Race and sex disparities in long-term survival of oral and oropharyngeal cancer in the United States. Journal of Cancer Research and Clinical Oncology, 142(2), 521–528. https://doi.org/10.1007/s00432-015-2061-8

Patrick, D. L., Lee, R. S. Y., Nucci, M., Grembowski, D., Jolles, C. Z., & Milgrom, P. (2006). Reducing oral health disparities: A focus on social and cultural determinants. BMC Oral Health, 6(1), 1-17. https://doi.org/10.1186/1472-6831-6-S1-S4

In addition, Center for Disease Control and Prevention (CDC) also use the term “disparities” to describe the racial/ethnic and socioeconomic groups’ differences in oral health, including oral health disparities in children, oral health disparities in adults, disparities in oral cancer and gum disease, and CDC’s work to reduce oral health disparities. (See Disparities in Oral Health, CDC, retrieved 03/30/2022 https://www.cdc.gov/oralhealth/oral_health_disparities/index.htm).

Based on the above information, we used the language “oral health disparities” in our article.

Line 45 - reformulated disadvantages with unfavorable socio-economic status.

Response: Thank you for the comments. We have revised the sentence accordingly to focus on socioeconomic status (now line 53).

Line 61 - reformulate communities of color - I suggest a single term of reference for the target population, for example African American people.

Response: Thank you for the suggestion. We have changed it to “African Americans” (see now line 62).

Line 66 - I suggest changing the term tooth pulled with tooth extracted.

Response: Thank you for the comments. We have changed the term to “tooth extracted” (now line 66).

The line containing lines 69-71 should be reformulated, it is not very well understood.

Response: Thank you for the comments. We have rewritten the sentence. It reads as “African Americans reported higher unmet dental care needs than their white peers (66.3% vs. 40.2%).” (now line 70).

Material and method

I think that this section could have some improvements, to be clearer.

Response: We’ve added a significant amount of additional discussion to the analytical section, which hopefully makes our methodology a lot clearer. See line 134-176.

Result

Table 1 - Rename Black with African American people or another term that you use everywhere.

Response: Thank you. We have renamed Black with “African American” in Table 1.

Figure 2 - Rename Percent Black to African American people or another term you use everywhere.

Response: Thank you. We have renamed Black with “African American” in Figure 2.

I suggest moving table 2 to Line 189

Response: Thank you. We had moved Table 2 as suggested.

I suggest that after Figure 3 appears in the text, Figure 3 should also appear.

Response: We mentioned Figure 3 in line 934. Figure 3 appear right after that.

Figure 3 is not clear why two graphs are the same but with different colors, please explain that it is not understood, possibly explanations on each graph called A and B or otherwise. 

Response: Thank you for the comment. We have revised Figure 3 so the distinction between the two charts is clearer. Figure 3 (a) is “Miles to Nearest Dental Services”; Figure 3 (b) is “Miles to Nearest Dental Services, including Federally Qualified Health Centers”.

Table 3 - line 234 change percent Black with the chosen term.

Response: Thank you. We have changed “Black” with “African American”.

Discussions - the discussions are comprehensive and to the point.

Response: We thank you for your encouraging comments.

Conclusions - I suggest removing the sentence with the bibliography -line 335 and reformulate.

Bibliography - Bibliographic sources must be entered in numbers with numbers in square brackets according to the MDPI bibliography guide.

Bibliography must be written in MDPI format, see format.

Response: Thank you for your comments. We have rewritten the bibliography in MDPI format, including sources entered in numbers with numbers in square brackets according to the MDPI bibliography guide.

Reviewer 3 Report

My only suggestion is to discuss the negative health outcomes from poor dental care, including sepsis, heart disease and more. The negative health outcomes go far beyond tooth loss. 

Given that African American men have higher rates of tobacco use, this too is correlated with poor oral health. I recommend that the authors make a note of this in their implications section. 

Author Response

Reviewer 3.

My only suggestion is to discuss the negative health outcomes from poor dental care, including sepsis, heart disease and more. The negative health outcomes go far beyond tooth loss. 

Response: Thank you for the suggestion. We added the following content on negative health outcomes from poor dental care, from line 39-45.

“The importance of oral health as a component of comprehensive health has been well documented. Untreated oral health problems can lead to difficulties in chewing food, pain, systemic infections, hospitalization, and in some cases, death [1,2]. Oral health is directly associated with different disease conditions and has close linkages to general health [3,4]. Dental diseases, such as tooth decay and gum disease, are linked to chronic health conditions, including poor nutrition, stroke, coronary heart diseases, diabetes, obesity, and kidney disease [5,6].  Locoregional infections of dental origin have the potential to cause sepsis [7]. Previous analyses have also showed statistically significant relationships between poor oral health and lost work hours, functional limitation, depression, and poor quality of life [8].”

We also added in “Conclusion” Line 351:

“Oral health access is an important social justice and health issue [14]. Routine and regular dental care not only prevents periodontal disease but also helps to screen patients who are at significant risk for more serious systemic conditions [15].”

Given that African American men have higher rates of tobacco use, this too is correlated with poor oral health. I recommend that the authors make a note of this in their implications section. 

Response: Thank you for your suggestion. We have address this in the “Implication” section, see line 339-345.

“Further, although cigarette smoking has substantially declined in the recent decades, there are still 19.4% African American adults currently smoking [55]. African American men are also at the highest risk of developing oral cancer [16]. Routine dental visits can examine oral cavity for signs of mucosal and tooth changes, recognize variations from normal, establish management and referral pathways [56]. Dental providers are also ideally positioned to provide brief interventions to assist their patients to quit smoking [56].”

We thank you very much for your time reviewing our study. Your valuable suggestions are greatly appreciated.

Reviewer 4 Report

  1. With regards to the title, perhaps stating the name of the state, and then country may better inform readers (e.g., Oral Health Service Access in Racial/Ethnic Minority Neighborhoods: A Geospatial Analysis in the District of Columbia, Washington D.C, U.S.A)
  2. Beware of lengthy sentences: lines 26-28- Changes are needed in health policies and programs, workforce development, resource allocation, community outreach, and educational programs in the provision of oral health care services for communities with higher percentages of low-income African American populations.
  3. Pay attention to spellings: line 35- chowing (chewing); line 43- toothy (tooth)
  4. A useful and applicable reference to enhance the discussion in lines 54-72: Patel, N., Patel, S., Cotti, E., Bardini, G. and Mannocci, F., 2019. “Unconscious Racial bias may affect dentists’ clinical decisions on tooth restorability: a randomized clinical trial”. JDR Clinical & Translational Research4(1), pp.19-28. https://doi.org/10.1177/2380084418812886
  5. Pay attention to sentence structures: line 91- contribute to the use…; lines 104-105- invested in the west of the river..
  6. A useful and applicable reference to strengthen the discussion in lines 318-322: Modha, B.(2022), "Utilising dentist-dental health educator skill-mix to implement oral health promotion that better supports diverse communities", Journal of Integrated Care, Vol. ahead-of-print No. ahead-of-print. https://doi.org/10.1108/JICA-08-2021-0043
  7. Overall, a well written paper that covers a very relevant and important issue that shall be of benefit to an international audience. The figures, references, and statistics are of satisfactory standard. The above two suggested references would make valuable additions; the second paper has been attached for your reference.
  8. Please ensure the language, grammar, punctuation, spelling and sentence structures are carefully assessed and refined to ensure a succinct and coherent read. This is because there are minor discrepancies in the language, grammar, punctuation, spelling and sentence structures. If need be, please obtain the necessary scientific English language reading and editing assistance, so that the paper has the potential to be read enjoyably by the international readership.

Author Response

Reviewer 4.

With regards to the title, perhaps stating the name of the state, and then country may better inform readers (e.g., Oral Health Service Access in Racial/Ethnic Minority Neighborhoods: A Geospatial Analysis in the District of Columbia, Washington D.C, U.S.A)

Response: Thank you for your suggestion. We have changed the title into “Oral Health Service Access in Racial/Ethnic Minority Neighborhoods: A Geospatial Analysis in Washington D.C, U.S.A”

Beware of lengthy sentences: lines 26-28- Changes are needed in health policies and programs, workforce development, resource allocation, community outreach, and educational programs in the provision of oral health care services for communities with higher percentages of low-income African American populations

Response: We have rewritten this sentence. See below:

“Improving access to racial/ethnic minority communities, especially African American neighborhoods, will required changes in health policies and programs, workforce development, resource allocation, community outreach, educational programs.”

Pay attention to spellings: line 35- chowing (chewing); line 43- toothy (tooth)

Response: Thank you very much. We have corrected such spellings.

A useful and applicable reference to enhance the discussion in lines 54-72: Patel, N., Patel, S., Cotti, E., Bardini, G. and Mannocci, F., 2019. “Unconscious Racial bias may affect dentists’ clinical decisions on tooth restorability: a randomized clinical trial”. JDR Clinical & Translational Research4(1), pp.19-28. https://doi.org/10.1177/2380084418812886

Response: Thank you for the suggesting the helpful reference. We had added the related content and reference to enhance the discussion.

“Further, effects of racism may also be a cause for differences in health outcomes that are examined. Implicit bias arising from subconscious thoughts or beliefs may also influence dental care providers’ approach, decision making, or patient management [21]. For instance, Patel et al. [22] found that a dentist’s decision making was affected by the race of the patient, resulting a greater likelihood of extractions for African American patients with a broken-down tooth and symptoms of irreversible pulpitis.”  

Pay attention to sentence structures: line 91- contribute to the use…; lines 104-105- invested in the west of the river.

Response: Thank you. We have made the revisions accordingly and conducted a thorough proof-reading for sentence structures and grammars.

A useful and applicable reference to strengthen the discussion in lines 318-322: Modha, B.(2022), "Utilising dentist-dental health educator skill-mix to implement oral health promotion that better supports diverse communities", Journal of Integrated Care, Vol. ahead-of-print No. ahead-of-print. https://doi.org/10.1108/JICA-08-2021-0043

Response: Thank you. We have added related content based on this reference. See line 1657-1661.

“Dental health educators can play a unique and valuable role in such community education and outreach programs through their interactions with patients, community members, and other stakeholders. They can collaborate with dental health providers in promoting oral health programs that better support racial and ethnic communities through effective interprofessional, skill-mix efforts [54].”

Overall, a well written paper that covers a very relevant and important issue that shall be of benefit to an international audience. The figures, references, and statistics are of satisfactory standard. The above two suggested references would make valuable additions; the second paper has been attached for your reference.

Thank you for your encouraging comments. We thank you for your suggestion. The suggested references enhanced the discussion of the article.

Please ensure the language, grammar, punctuation, spelling and sentence structures are carefully assessed and refined to ensure a succinct and coherent read. This is because there are minor discrepancies in the language, grammar, punctuation, spelling and sentence structures. If need be, please obtain the necessary scientific English language reading and editing assistance, so that the paper has the potential to be read enjoyably by the international readership.

Response: Thank you. We have conducted a thorough proof-reading to ensure the language and grammar to ensure a succinct and coherent read.

We thank you very much for your time reviewing our study. Your valuable suggestions helped us to improve our article and are greatly appreciated.

Reviewer 5 Report

The authors in this study have provided a geospatial analysis in the district of Columbia with a focus on oral health access in racial/ethnic minority neighborhoods.

Importance of the study: This study would allow understanding the unequal and inequitable distribution of the burden of oral diseases and risks, based on individual-level variables.

The overall impact of this study is to improve our knowledge base of oral health service use, environmental or contextual variables that are important in oral health disparities in racial and ethnic neighborhoods.

It is interesting to know that the urban areas with highly diverse populations like Washington DC, racial disparities in oral health care access are significant. The results clearly indicate that higher concentrations of African Americans are associated with decreased oral health care access, irrespective of age, gender, and access to Federal Quality Health Centers.

Though the study is centered on only one ethnic group (limitation of the study), however, it does provide important information that points towards the need for changes in health policies and programs, workforce development, resource allocation, community outreach, educational programs in the provision of oral health care services for communities with higher percentages of low-income African American populations.

Minor drawbacks: 1) The methodology can be elaborated to understand clearly the different analysis that was applied, for example, the QGIS.

2) Educational status and lack of awareness about oral health among such population could be the factors to consider as well in this study.

3) Addition of another ethnic group for comparison.

Overall, it is a good study with a thorough analysis.

Author Response

Reviewer 5.

The authors in this study have provided a geospatial analysis in the district of Columbia with a focus on oral health access in racial/ethnic minority neighborhoods.

Importance of the study: This study would allow understanding the unequal and inequitable distribution of the burden of oral diseases and risks, based on individual-level variables.

The overall impact of this study is to improve our knowledge base of oral health service use, environmental or contextual variables that are important in oral health disparities in racial and ethnic neighborhoods.

It is interesting to know that the urban areas with highly diverse populations like Washington DC, racial disparities in oral health care access are significant. The results clearly indicate that higher concentrations of African Americans are associated with decreased oral health care access, irrespective of age, gender, and access to Federal Quality Health Centers.

Though the study is centered on only one ethnic group (limitation of the study), however, it does provide important information that points towards the need for changes in health policies and programs, workforce development, resource allocation, community outreach, educational programs in the provision of oral health care services for communities with higher percentages of low-income African American populations.

Minor drawbacks: 1) The methodology can be elaborated to understand clearly the different analysis that was applied, for example, the QGIS.

Response: Thank you. We agree with the reviewer. We have elaborated the methodology section and clarified the analysis we used, including GQIS.  

2) Educational status and lack of awareness about oral health among such population could be the factors to consider as well in this study.

Response: We included this in our implication section. “Previous studies have shown lower educational attainment is associated with decreased oral health literacy levels and heightened the treat to oral health [49,50]. In addition, a lack of awareness around oral health is also a significant barrier [51].” (Line 325-327).

3) Addition of another ethnic group for comparison.

Response: Thank you for your suggestion.

We included neighborhood tract-level data on race/ethnicity (African American and Hispanic) in our analysis, see Figure 2, Table 2, and Table 3. We found that “The Hispanic/Latino communities seem to be most concentrated in Wards 1 and 4, as well as Ward 3” (Line 200); neighborhoods with a “higher percentage of Hispanic residents and higher median ages were associated with shorter distances to their nearest oral health providers.” (Line 222). In addition, “census tracts with higher concentrations of Hispanics were associated with decreased distances to oral health providers (β=-0.61, p<0.05).” (Line 242)

Overall, it is a good study with a thorough analysis.

We thank you again for reviewing our study. Your valuable suggestions are greatly appreciated.

Round 2

Reviewer 4 Report

Following the authors efforts in making additions and changes to the manuscript, the manuscript has reached a standard where publication can be considered. Once the authors have revisited their paper, thoroughly assessed and polished it to ensure a succinct and coherent read, I would expect the paper to be in good stead for publication. I would deem the paper to be: Accept after Minor Revision, as minor text editing may be required.

Author Response

Dear Reviewer,

Thank you for your feedbacks. Three of the authors have thoroughly assessed and revised the manuscript for text editing to ensure a succinct and coherent read. Again we really appreciate your valuable time reviewing our manuscript.

Please see the attached revised manuscript. Please let us know if there are further questions. Thank you. 
